# MODEL EXTRACTION ATTACKS ON DISTILBERT

## ABSTRACT

This paper investigates model extraction attacks, where an adversary can train a substitute model by collecting data through query access to a victim model and steal its functionality. We use DistilBERT as the victim model due to its smaller size and faster processing speed. The results demonstrate the effectiveness of the model extraction attack and show that fine-tuning more powerful language models can improve accuracy. The study provides important insights into the security of machine learning models.

## 1 INTRODUCTION

A Model Extraction Attack -shown in figure 1- is when an adversary can collect data through query access to a victim model and train a substitute model with it in order to steal the functionality of the target model (Krishna et al., 2019; Keskar et al., 2020; He et al., 2021).

DistilBERT (Sanh et al., 2019) is a small, fast, cheap, and light Transformer model (Vaswani et al., 2017) based on the BERT architecture Devlin et al. (2018). We use DistilBERT as the victim model because it is smaller and faster than the `BERT-base` model, and retains most of its functionality. Hence, DistilBERT is cheaper to deploy, and using it as a victim model will be more realistic.

## 2 METHODOLOGY

For this study, we employ extraction attacks by training two DistilBERT victim models. One of these models is trained on the Ag News dataset (Zhang et al., 2015), while the other is trained on the TrustPilot Reviews dataset (Hovy et al., 2015).

We conduct experiments under two scenarios. In the first scenario, we assume the attacker knows that the victim model is a DistilBERT model, so we train three extracted models based on DistilBERT using a different dataset for each one of them. The first dataset is the same dataset that is used to train the victim model with the original labels replaced by the predictions of the victim model ($D_A = D_V$) and the second one is a dataset from the same domain with the same size as the training dataset with labels extracted from the victim model ($D_A \neq D_V$, 1x), and the last one is a dataset from the same domain with a size equal to five times the size of the training dataset with labels extracted from the victim model ($D_A \neq D_V$, 5x).

In the second scenario, we assume the attacker does not know the victim model, so we test four models with different sizes, `BERT-base`, `BERT-small` (Turc et al., 2019), `TinyBert` (Jiao et al., 2019), and a model with the same architecture as `DistilBERT` but trained from scratch (`BERT-base > DistilBERT > BERT-small > TinyBert`).

It should be noted that we use the Yelp polarity (Zhang et al., 2015) to extract the model trained on the reviews' dataset and Yahoo Answers (Zhang & Lecun, 2015) to extract the model trained on the news' dataset. The details of our experimental setup can be found in Appendix A.

## 3 RESUTLS

Table 1 shows the results. We report the accuracy and loyalty of each model. Loyalty is the accuracy while using victim predictions as reference labels, it represents a measure of how successful is our extracted model in mimicking the behavior of the victim model. The Table contains multiple blocks. The victim block represents the results of the victim model, while each block from the rest of the blocks represents the results of the corresponding model as the extracted model.

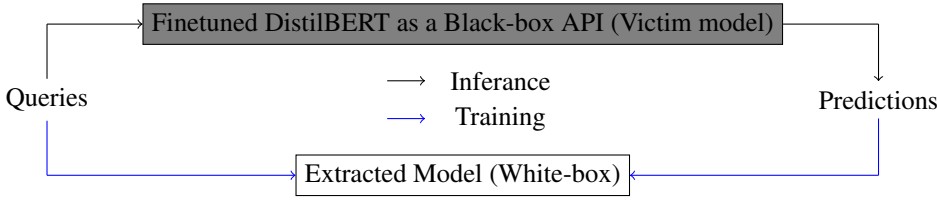

Figure 1: Model Extraction Attack: We pass queries to the victim model and use its predictions with the queries to train the extracted model.

Table 1: Accuracy and Loyalty of the victim models and the extracted models among different datasets. #Q is the number of queries.

| Model | #Q | AG News | | Trustpilot reviews | |
|---|---|---|---|---|---|
| | | Accuracy | Loyalty | Accuracy | Loyalty |
| **Victim** | | 93.91% | - | 88.00% | - |
| **DistilBERT** | | | | | |
| $D_A = D_V$ | | 94.58% | 98.64% | 88.20% | 95.20% |
| $D_A \neq D_V$ | 1x | 89.96% | 90.64% | 87.60% | 91.40% |
| $D_A \neq D_V$ | 5x | 91.88% | 92.21% | 87.97% | 93.40% |
| **BERT-base** | | | | | |
| $D_A = D_V$ | | 94.92% | 98.08% | 88.70% | 94.20% |
| $D_A \neq D_V$ | 1x | 90.64% | 88.27% | 87.10% | 91.30% |
| **BERT-small** | | | | | |
| $D_A = D_V$ | | 93.01% | 97.74% | 87.50% | 93.30% |
| $D_A \neq D_V$ | 1x | 88.95% | 86.69% | 85.00% | 89.50% |
| **TinyBERT** | | | | | |
| $D_A = D_V$ | | 93.46% | 95.94% | 88.60% | 93.70% |
| $D_A \neq D_V$ | 1x | 85.56% | 85.79% | 86.40% | 90.30% |
| **DistilBERT from scratch** | | | | | |
| $D_A = D_V$ | | 92.44% | 93.79% | 87.70% | 91.60% |
| $D_A \neq D_V$ | 1x | 66.62% | 79.93% | 80.40% | 82.80% |

In all our models used for extraction, we can see that the accuracy and the loyalty of the extracted model are higher when $D_A = D_V$, which indicates that the success of model extraction depends on the similarity between the training data of the victim model and the attacker's queries.

Overall, for pre-trained extracted models, we observe that more models' parameters lead to higher accuracy. The only exceptions are BERT-small and TinyBERT. Although TinyBERT has fewer parameters, it extracts the victim better than BERT-small. We argue that this is because TinyBERT was trained with Knowledge Distillation, similar to the victim (DistilBERT).

Also, the DistilBERT extracted models trained with 5 times the size of the training dataset (5x) outperforms the ones trained with the same size of the training dataset (1x), indicating that increasing the number of queries leads to improved extraction performance.

Finally, The extracted models based on pre-trained DistilBERT have higher accuracy and loyalty compared to the corresponding extracted models that are trained from scratch, emphasizing the importance of using pre-trained models.

## 4 CONCLUSION

Our study provides a comprehensive analysis of extraction attacks on DistilBERT models, revealing potential vulnerabilities associated with these models. Furthermore, The results obtained have significant implications for the security of DistilBERT models and inform the development of more robust defense mechanisms against such attacks. Overall, our study contributes to the growing body of research on adversarial attacks in Natural Language Processing and underscores the importance of continued investigation into model security.

URM STATEMENT

The authors acknowledge that at least one key author of this work meets the URM criteria of the ICLR 2023 Tiny Papers Track.

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

# A EXPERIMENTAL SETUP

## A.1 DATASETS

**Trustpilot Sentiment dataset** (Hovy et al., 2015): It contains reviews associated with a sentiment score on a five-point scale. In this paper, we only use two scores, score one as a negative review and score five as a positive review.

**Yelp Polarity dataset** (Zhang et al., 2015): Yelp dataset is a document-level sentiment classification dataset. The original dataset is on a five-point scale, while the polarised version assigns negative labels to the rating of 1 and 2 and assigns positive labels to 4 and 5.

**AG News dataset** (Zhang et al., 2015): AG news corpus is a dataset that's mainly used for topic classification task which is to predict the topic label of the document. It has four different topics in total. We only used three topics: "Sports", "Business" and "Sci/Tech" for our research purposes.

**Yahoo Answers Topics dataset** (Zhang & Lecun, 2015):Yahoo answers topics covers 10 different

| Dataset | Train | Validation | Test | Task |
|---|---|---|---|---|
| Trustpilot | 22,142 | 2,767 | 2,767 | sentiment analysis |
| Yelp Polarity | 520K | - | - | sentiment analysis |
| AG News | 7,100 | 887 | 887 | topic classification |
| Yahoo Answers | 590K | - | - | topic classification |

Table 2: Statistic of sentiment analysis and topic classification datasets.

topics.however, We used only three topics of them for our studies: "sports", "Business & Finance" and "Science & Mathematics"
These datasets are shown in table 2.

## A.2  MODELS

Our research study focuses on developing a victim model that is based on fine-tuning a DistilBERT model Sanh et al. (2019) for a sequence classification task.

To extract information from the victim model, we employed two different approaches:
The first approach involved training the attacker multiple times by fine-tuning several models including DistilBERT Sanh et al. (2019), BERT-base Devlin et al. (2018), BERT-small Turc et al. (2019), and TinyBERT Jiao et al. (2019). This approach allowed us to observe the behavior of the attacker model when it is pretrained and have a prior understanding of the language.
The second approach we employed was to train a DistilBERT model from scratch using the extracted dataset. This approach helped us to gain insights into the performance of a new model on the same task and evaluate its effectiveness in performing attacks

## A.3  HYPER-PARAMETERS

We finetune all models for $4$ epochs with a learning rate of $5e-5$, and a batch size of $16$ example per batch. This study employed Huggingface libraries[1] for training purposes.

---

[1]https://huggingface.co/

