# OpenReview forum: "Model Extraction Attacks on DistilBERT"
_ICLR.cc/2023/TinyPapers — Submitted to Tiny Papers @ ICLR 2023_

### Official Review · Reviewer_d4KG · 2023-03-27

**Confidence:** 4

**Summary Of Contributions:**

The paper considers the model extraction attack problem in the context of language models. Experiments are provided to compare accuracy and loyalty of attacks, on reviews and news data sets.

**Rating:**

Needs Clarification (NC): a submission which does not meet the reviewing criteria and needs clarification for its described problem or solution

**Strengths And Weaknesses:**

- Interesting problem direction.
- Needs some clarification (see detailed suggestions below).

**Suggested Changes:**

Clarity:
- Terms $D_A$ and $D_V$ are not defined, would be good to state explicit definition for the variables (are they attacker/victim domain? or data sets?).
- Abstract and introduction need to be more carefully written: in particular, include stronger coupling to current contributions; motivation and context for the model extraction attack problem; with more clarity for any claims made (e.g. why cheapness of model makes it a more realistic victim model? more realistic compared to what?)

Correctness:
- Need to clarify that how the hyperparameters compare to original DistilBERT training E.g. Was there any grid search done, or original hyperparameters were reused (note that replication of models by copying original parameters is not ideal). It is hard to determine correctness of experiments without this information.
- The claim "show that fine-tuning more powerful language models can improve accuracy" is not connected to provided experiments, and is recommended to be removed from the abstract.

Reproducibility:
- code used should be linked to (with appropriate anonymization).
- missing details: optimizer; learning rate schedule; regularization.

Additional suggestion, to potentially make work more comprehensive:
- Would be good to try to investigate and explain the following:
-- Table 1: Why is the attacker accuracy better than victim model for DistilBERT? Accuracy of 'DistilBERT from scratch' seems too low on AG News - why?
- Why mimicking only via transformers? What if the attacker does not know the exact architecture used in the victim model? What if the victim model was another sequence classification model?

Also, the abstract in the open-review seems to be incorrectly uploaded?

---

### Official Review · Reviewer_mCq7 · 2023-04-01

**Confidence:** 4

**Summary Of Contributions:**

The paper provides a comprehensive analysis of model extraction attacks on DistilBERT. The authors do so by querying two victim models in different adversarial settings.

**Rating:**

Great Start (GS): a submission which meets some of the reviewing criteria but has room for improvement

**Strengths And Weaknesses:**

**Strengths**:

- The paper studies an essential and timely problem.
- I appreciate the authors taking the time to add supplementary information about the experimental setup and the model training workflow.

**Weaknesses**:

- Given the motivation of the Tiny Papers track, reproducibility is paramount. The absence of code is a major weakness. I hope the authors look to add it later.
- The paper does miss out on some important references [1,2] which perform similar studies.
````
[1] He, X., Chen, C., Lyu, L., & Xu, Q. (2022). Extracted BERT Model Leaks More Information than You Think!. arXiv preprint arXiv:2210.11735.

[2] Chen, C., He, X., Lyu, L., & Wu, F. (2021). Killing One Bird with Two Stones: Model Extraction and Attribute Inference Attacks against BERT-based APIs. arXiv preprint arXiv:2105.10909.
````

**Suggested Changes:**

Overall a solid paper, but I have some suggestions to improve the reading experience for others.

- The illustration of the framework used by the authors could be improved. In its current state, it isn't easy to get an overview of the experimental setup by just looking at the picture.
- While the results in Table 1 make sense, the actual Results section (Section 3) presents it somewhat haphazardly. It took me some back-and-forth to correlate the text with the values in the table. The authors should try and improve the flow of the section. The same holds true for some of the other sections as well.
- I understand the constraint of space, but I would recommend touching on papers that deal with similar querying attacks - to help readers gather further context (listed out in `Weaknesses`).

(Edit): After submitting the review, I also noticed that the Abstract seems to be uploaded weirdly in OpenReview.

---

### Official Review · Reviewer_GxR8 · 2023-04-03

**Confidence:** 5

**Summary Of Contributions:**

This paper implement extraction attacks to DistilBERT that is trained on two different dataset.

**Rating:**

Needs Clarification (NC): a submission which does not meet the reviewing criteria and needs clarification for its described problem or solution

**Strengths And Weaknesses:**

## Strengths:
1. The paper is easy to read.

## Weaknesses:
### 1. The conclusions of the paper are not surprising:
1) "More model parameters lead to higher accuracy." This conclusion is obvious, and nothing new is learned from the provided experimental results.
2) " The extracted models based on pre-trained DitsillBERT have higher accuracy and loyalty compared to the corresponding extracted models that are trained from scratch." This is also apparent in the existing literature.
### 2. The writing of this paper is problematic. This is a rash submission.
1) There is even **a typo in the title**: 'DitsilBERT' should be 'DistilBERT'. And the format of the abstract on the submission page is not right. There is no space between the words.
The author should at least use their own expression to write the paper because this Tiny Paper only needs a few sentences to explain the definition. For example, **the first sentence** of the  Introduction Section, "A Model Extraction Attack -shown in figure 1- is when an adversary can collect data through query
access to a victim model and train a substitute model with it in order to steal the functionality of the
target model" **is way too similar to a sentence from the abstract of an existing paper** "The definition of model extraction attack is that an adversary can collect data through query access to a victim model and train a substitute model with it in order to steal the functionality of the target model." [1]
### 3. There is no novelty in the idea.
The contribution of the paper is only to implement a model extraction attack to DistilBERT, so it's not a novel idea.

Reference:
[1] Liu, Shengyi. "Model Extraction Attack and Defense on Deep Generative Models." In Journal of Physics: Conference Series, vol. 2189, no. 1, p. 012024. IOP Publishing, 2022.

**Suggested Changes:**

1. Should have a chart summarizing the number of parameters of different models.
2. The writing of this paper should be largely improved.
3. Should re-design the experiments for a clear goal. The current results do not provide any insights or constructive opinions.

---

### Meta-Review · Area_Chair_WEfy · 2023-04-07

**Recommendation:** Invite to archive
**Confidence:** 3

**Metareview:**

Thank you for your submission. As the reviewers have noted, this is an interesting and timely problem. Overall, they felt that this paper was a great start, but could be improved in the following ways: make code available for better reproducibility, add discussion on closely related work (papers mentioned in the reviews), additional details provided on training and experiments, and some added discussion on the intuition of the various results.

**Summary:**

This paper investigates model extraction attacks on DistilBERT.

**Reason For Not Giving A Higher Recommendation:**

- Missing some necessary related work
- Code not provided
- Requires some editing (symbols defined, supporting claims, etc.)
- Clarifications needed on experimental setup and discussion of results

**Reason For Not Giving A Lower Recommendation:**

- Interesting problem
- Thorough experimentation

---

### Decision · Program_Chairs · 2023-04-09

**Decision:**

Invite to archive

**Comment:**

Please reformat your abstract on OpenReview so it shows correctly.